# Barriers and facilitators of severe acute malnutrition management at Felege Hiwot Comprehensive Specialized Hospital, Bahir Dar, North West Ethiopia, descriptive phenomenological study

**Daniel Alelign[1], Netsanet Fentahun[2], Zeamanuel Anteneh Yigzaw[3]***

1 Department of Nursing, Felege Hiwot Comprehensive Specialized Hospital, Amhara Regional Health Bureau, Bahir Dar, Ethiopia, 2 Department of Nutrition and Dietetics, School of Public Health, College of Medicine and Health Sciences, Bahir Dar University, Bahir Dar, Ethiopia, 3 Department of Health Promotion and Behavioral Sciences, School of Public Health, College of Medicine and Health Sciences, Bahir Dar University, Bahir Dar, Ethiopia

* zeamanuel19@gmail.com

**Data Availability Statement:** All relevant data are within the paper and its supporting information files.

## Abstract

### Background

Malnutrition is a clinical condition that affects all age groups, and it remains a major public health threat in Sub-Saharan Africa. As a result, this research aimed to investigate the barriers and facilitators of treating severe acute malnutrition at Felege Hiwot Comprehensive Specialized Hospital in Bahir Dar City, North West Ethiopia.

### Methods

A descriptive phenomenological study was conducted from February to April 2021. The final sample size taken was fifteen based on data saturation. In-depth and key informant interviews were conducted with nine caregivers, three healthcare workers, and three healthcare managers supported by observation. A criterion-based, heterogeneous purposive sampling technique was used to select the study participants. Each interview was audio-taped to ensure data quality. Thematic analysis was done to analyze the data using Atlas. ti version 7 software.

### Results

Two major themes and six sub-themes emerged. Barriers related to severe acute malnutrition management include subthemes on socio-economic and socio-cultural conditions, perceived causes of severe acute malnutrition and its management, and the healthcare context. Facilitators of severe acute malnutrition management include severe acute malnutrition identification, service delivery, and being a member of community-based health insurance.

**Funding:** The authors received no specific funding for this work.

**Competing interests:** The authors have declared that no competing interests exist.

**Abbreviations:** CF, Complimentary Feeding; CMAM, Community-Based Acute Malnutrition Management; CASH, Clean And Safe Hospital); CRC, Compassionate respectful care; ETB, Ethiopian Birr); FHCSH, Felege Hiwot Comprehensive Specialized Hospital; GMP, (Growth Monitoring Program; HCW, Health care worker; IDI, In-depth interview); KII, key informant interview); MAM, Moderate Acute Malnutrition; MUAC, Middle Upper Arm Circumference; RUTF, Ready to Use Therapeutic Food; SAM, Severe Acute Malnutrition; SC, Stabilization center; STG, Standard treatment guideline; U5, Under five; UN, United Nation; WHO, World Health Organization.

## Conclusions

Effective management of severe acute malnutrition was affected by a multiplicity of factors. The results reaffirm how socioeconomic and sociocultural conditions, perceived causes of severe acute malnutrition (SAM) and its management and the health care context were the major barriers, while able to identifying severe acute malnutrition, service delivery, and is a member of community-based health insurance were the major facilitators for SAM management. Therefore, special attention shall be given to SAM management.

## Introduction

The World Health Organization (WHO) defined malnutrition as a cellular imbalance between the availability of nutrients and energy and the body's demand for them to ensure growth, maintenance, and specific functions. Malnutrition remains a global health concern and contributes significantly to childhood mortality. Nearly half of all deaths in children <5 years of age are attributed to undernutrition, especially in developing countries [1].

Severe acute malnutrition (SAM) in under-5-year-old children is a neglected tropical disease (NTD) and a non-communicable disease (NCD) among low-income, impoverished communities with the highest risk of dying [2–4]. Nearly half of all deaths worldwide among children under 5 years of age are linked to undernutrition. Nutritional well-being during this critical period has both immediate and lasting consequences on a child's physical and cognitive health, development, and functioning [5].

Evidences verifying that different understandings of physiological states exist and vary between cultures has been well described by medical anthropologists, as have the many other social and cultural factors that influence health-seeking behavior. However, an in-depth understanding of community perceptions and social aspects of undernutrition and community-based management of acute malnutrition (CMAM) was absent in this context, which placed a substantial barrier to effective and sustainable program implementation [6,7]. There have been identified challenges or barriers to treat severe acute malnutrition, which range from individual, community, institutional, and multi-sectorial nutrition involvement and engagement [8].

Malnutrition is a top public health problem in low- and middle-income countries. Worldwide, there are about 165 million stunted, 52 million wasted, and 17 million severely wasted under-five children. Globally, malnutrition is the underlying explanation for the death of about 3 million (45%) under-five children per year. One of these 11.6% (804,000) children dies due to suboptimal breastfeeding. Strong scientific evidence suggests that insufficient quantities and inadequate quality of complementary foods (CFs), poor child-feeding practices, and a heavy burden of infectious illnesses have adverse effects on child survival, growth, and development. Poverty, food insecurity, ignorance, poor hygiene, and poor sanitation are some other factors responsible for high levels of child malnutrition in developing countries [9–11].

Nutrition has a substantial role not only for children's, it is also important for the prevention, treatment, and cure of many diseases for every person including tuberculosis [12]. Many disease-related malnutrition is also a challenge among hospitalized patients [13]. Acute malnutrition, or wasting, is an attributable cause of 12.6% of the 6.9 million deaths among children under 5 years old, accounting for more than 800,000 deaths annually [14,15].

The effective implementation of large-scale nutrition interventions in Africa is an ongoing challenge [16]. The prevalence of undernutrition in children under 5 years of age remains particularly high in Africa and South East Asia, close to 35% of the 7.6 million deaths that occur

each year among children who are under 5 years of age because of nutrition-related factors, and about such deaths are specifically attributable to severe wasting [17]. Severe malnutrition is one of the common causes of morbidity and mortality among children under the age of 5 years worldwide. Based on the 2019 Ethiopia Demographic Health Survey (EDHS), 7% of children under five years old are wasted, and 1% of these are severely wasted. Children with severe acute malnutrition (SAM) have a nine-fold increased risk of mortality compared to well-nourished children [4,18]. Cultural and traditional feeding practices influence the type, quantity, and frequency of food given to women and children [9].

In general, research has shown that socio-cultural practices and perceptions play an important role in SAM management. Therefore, exploring the barriers and facilitators for the management of SAM among mothers/caregivers, healthcare providers, and healthcare managers will have a great role in improving the management of SAM at all levels.

## Methods

### Study design

A descriptive phenomenological study design was used from February to April 2021.

Phenomenological study design is used to attain the lived experience and an understanding of the right meaning of a phenomenon of interest through engaging detailed descriptions of the mothers or caregivers and service providers' perception toward barriers and facilitators during managing severe acute malnutrition [19–21].

### Study setting

The study was conducted in Bahir Dar city, Amhara Regional State, North-West Ethiopia. Bahir Dar is the capital city of Amhara Regional State. The city was found at 1801 meters above sea level and 565 kilometers from Addis Ababa, which is the capital city of Ethiopia. The city has three governmental and four private hospitals, and Felege Hiwot Comprehensive Specialized Hospital (FHCSH) has a better-structured severe acute malnutrition center. According to the FHCSH 2019/2020 report, the hospital provides SAM management services per year for 236 SAM patients from different districts of the Amhara region and neighboring regions.

### Study participants

Caregivers/mothers aged 18 years and older with children under five years of age who are undertaking SAM treatment.

Healthcare workers who work at SAM ward.

Health care managers or coordinators of the hospital.

**Sample size determination.** The final sample size taken was fifteen (15), nine care providers, three healthcare managers, and three healthcare providers. The number of participants recruited was based on data saturation, that is data were collected until no new themes were identified and no issues emerged from the interviews. A criterion-based, heterogeneous purposive sampling technique was used to select the study participants.

Most qualitative researches describe the sample size for qualitative research was dependon data saturation more over a known qualitative scientist Creswell suggests that a reasonable sample size may range from 3 to 25 participants forphenomenological study design [22].

**Data collection procedure and tool.** Data for this study was gathered by using an in-depth interview and key informant interview guide, and observation guides of the facility and clinical practices of health care providers (S1 File). Data collection was taken care of by the principal investigator, and notes were taken (to capture memos). Data collection started with

an observation of the participants in the SC (stabilization center) unit, and the interview was initiated with a broad and general question followed by a probing question, and then the questions got more focused as the data collection progressed. The participants were engaged intrinsically in a private and conducive room during data collection; the process began by first asking the participant and listening attentively until ideas were complete, and then probing was done based on the participant's response. Each interview was audiotaped using a digital voice recorder.

**Data analysis and processing.** Each interview was audiotaped, transcribed word from the audio tape to an Amharic transcript, and then translated to English by the principal investigator. Every day, after completing each interview, the field notes taken from the observation were rewritten, including the participant's nonverbal communication, and they listened to the recorded document several times. Thematic analysis was done to analyze the data.

The data was managed and coded using Atlas Ti version 7 software, 2016 after the investigator read the transcript line by line at different times. Data with similar meanings and concepts formed sub-themes, which then evolved into major themes.

**The rigor or trustworthiness of the study.** The trustworthiness of this study was ensured through transferability, dependability, credibility, and confirmability principles.

## Transferability

The transferability of the study was increased by describing the study setting, sample, and data collection procedure clearly and in detail. Thick descriptions were provided for the methodological procedures and interpretation of the results. We have done purposive sampling to get specific data relative to the context. All interviews were conducted via audio recording, and notes were taken for observational data.

## Dependability

The dependability of the study was obtained through prolonged and friendly engagement with participants to develop trust in the interviewer. With the help of the supervisor, the research team created a data audit to ensure that the data and findings were rich-thick, consistent, and stable over time. Furthermore, the findings are reported with supportive quotations which opened a door for the reader to evaluate and build trust in the interpretations. In addition, the findings were audited and verified by qualitative research experts.

## Credibility

The credibility of the study was maintained by triangulating interview data with observations. The interview guide and observation checklist were evaluated by professionals before data collection. To accomplish this, we have spent time in the field, and done peer debriefing and member checks. To obtain an in-depth understanding, it has been kept throughout the interview.

## Conformability

The conformability of the study was proved by using data audit and triangulation through detailed recording of every activity of the participant during the time of the interview and every procedure that was done throughout the study.

## Ethics approval and consent to participate

Ethical clearance was obtained from the institutional review board (IRB) of the College of Medicine and Health Sciences of Bahir Dar University. The ethical letter number allowed to

**Table 1. Socio-demographic characteristics of the study participants.**

| | Mothers / caregivers | Health care Workers (HCW) | Healthcare managers/coordinators (HCM) |
|---|---|---|---|
| Type | IDIs | KII | KII |
| **Age** | | | |
| Mean | 33 | 34 | 32 |
| 22–32 | 7 | - | 1 |
| 33–41 | 1 | 3 | 2 |
| > 42 and above | 1 | - | - |
| **Educational status** | | | |
| No formal education | 4 | - | - |
| Primary 1–8 grades | 2 | - | - |
| Secondary 9–12 grades | 1 | | |
| Tertiary and above | 2 | 3 | 3 |
| **Residence** | | | |
| Urban | 3 | 3 | 3 |
| Rural | 6 | - | - |

conduct this research was a ref. no. ሜዲ. 11459/1.4.4, written on March 10, 2021. Informed consent was obtained both orally and in writing.

## Results

### Socio-demographic characteristics of the study participants

Fifteen individuals were interviewed. These include nine caregivers, three healthcare providers, and three healthcare managers or leaders. Among those most of the caregivers of the child were aged greater than thirty. Six caregivers were from rural areas (Table 1).

**Thematic findings.** In this study, two major themes and six subthemes were identified (Table 2).

### Major Theme 1: Barriers related to severe acute malnutrition management

**Sub-theme 1: Socio-economic and socio-cultural conditions (views, preferences, care-seeking behavior, and access).** Most respondents said that treatment facilities were not available in their localities; they were not aware of home-based treatment options. All were aware that treatment could be obtained in hospitals.

Many respondents feel helpless when they suspect that their child is not growing adequately. Maternal nutrition and well-being were viewed as essential elements for the child's well-being. Some respondents also shared that they visited local traditional healers since they thought the cause of the illness could be an evil eye (**"Buda" or "yesewayin wogtot"**(ቡዳ/የሰው ዓይን ወግቶት).

> "*I preferred to give my child soup and cow's milk while he cried, and I gave him some traditional medicine to alleviate this problem, but it did not cure because I strongly suspected it was an evil eye (የሰው ዓይን ወግቶት(ቡዳ))*". (65 years old grandmother, IDI)

Almost all caregivers and key informants considered a community-based treatment for SAM to be a positive option that would be cheaper and more convenient than inpatient care. Moreover, they also raised some concerns and made associated suggestions. For example,

**Table 2. Thematic findings of the barriers and facilitators of SAM management.**

| Major Themes | Sub Themes | Concepts |
|---|---|---|
| **Major theme 1:**—Barriers to SAM management | Sub-theme 1:—Socio-economic and socio-cultural conditions (views, preferences, care-seeking behavior, and access) | Unavailability of treatment centers Perceiving as Evil Eye/ Budda Increased workload Staff inadequacy Delay coming for treatment |
| | Sub-theme 2:—Perceived causes of SAM and its management | Lack of knowledge Illiteracy Poor breast-feeding practice |
| | Sub-theme 3:—The health care context (logistics and trained manpower) | Logistics problem Lack of training |
| **Major theme 2:**—Facilitators of SAM management | Sub theme 1:—Able to identify SAM | Naming as Amenmin Describing as skinny or big-head Able to identify the problem SAM |
| | Sub theme 2:—Service delivery | Good service delivery Healthcare worker's friendly approach Facility layout |
| | Sub-theme 3:—Participation in community-based health insurance | Being a member of CBHI |

many said that health extension workers were not adequately trained to diagnose or provide quality treatment for critical conditions such as SAM. In addition, they proposed that treatment facilities need to be opened in their locality, which would reduce the cost of the treatment. The same suggestion was also made by several HCWs and healthcare managers.

If a child's condition worsens, hospital services are very expensive, both to access (e.g., transport costs to get to the hospital) and to stay (e.g., cost of diagnostics, cost of medicines, cost of beds, and cost of foods for caregivers). Other factors against long periods of inpatient care were the inability of the mother to continue with household chores, the need to care for other members of the family, and a lack of money to pay for services given to the child.

"*We stay for about one month here, and since we are all suffering from a financial crisis and cannot pay beyond this, it is a must to default on treatment and go home to take care of ourselves at home*". (34 years old caregiver mother, IDI)

"*Because the cost is getting bigger and people like us are not solvent enough to spend such a big amount of money. However, we would say it is better to treat such babies in the hospital.* (31 years old mother, IDI)

Health care providers (Doctors and nurses) in the malnutrition ward in this study area discussed some practical issues. They mentioned that the increased workload, while admissions increased to the malnutrition ward, affected the quality of care; staff adequacy is not always enough to provide the care they want to provide. Moreover, parents often present their children at a very late stage of illness, SAM can be complicated to manage. A strong community-

based model of care would identify cases earlier and thus help reduce mortality, as well as public and out-of-pocket expenditure. Although respondents believed that inpatient treatment is the best option for managing critical cases, all were also in favor of having community-based care options. Few were confident that mothers alone could take care of their babies, even if they were educated to provide home-based care.

Moreover, they suggest giving training to community HCWs to support mothers, making doctors/nurses more available, and providing equipment and medicine at the community clinics so that people need not go far for quality care of their sick babies:

"*We won't have to come to the hospital then, and it will save us money." There will be less tension, and I won't have to ask anyone for money. "If the baby can be treated in the locality, then the family members can also do their household chores properly.*" (30 years old caregiver mother, IDI)

Health Care Workers (HCWs) and Health Care Managers (HCMs) also mentioned the need to train and raise awareness of community leaders, such as religious leaders, to identify and support mothers with malnourished children. Other concerned people also need training.

A healthcare manager suggests that "*A local leader can educate the community about SAM if he receives training in the subject since the people in the area respect and believe him. Other concerned members of the community include teachers, who, with the right training, can assist children with SAM*". (35 years old health care manager, KII)

"*My children were left alone and starved while I was here; livestock was causing me problems; it was cropping cultivation season, and I couldn't work on time." This may exacerbate food insecurity; I have borrowed money for transportation and to purchase medicines that are not available in this hospital, so the financial problem is the primary burden for the entire family.*" (32 years old caregiver, IDI).

Most respondents identified mothers as having primary caring responsibility for a sick infant. Mothers were said to be best placed to determine the reason why their children were ill and to care for them at home. Fathers and in-laws were also said to be crucial in cases of severe illness when an infant needs to be taken to the hospital. Some also mentioned that fathers and other family members could help the mother while she is taking care of the sick baby at home. The mother bears the primary responsibility; she must inform the father of the baby's problem as well as her in-laws, but the mother will observe it first.

Most respondents in the family felt that HCWs can provide advice on what to do at home, follow up on improvement or deterioration, and refer to higher-level health and institutions if needed.

"*Influential and well-off members of the community could help caregivers, especially by providing financial or in-kind support to the family. HCWs supporting infants with SAM were expected to have "experience" with the condition and to be "helpful" with good intentions. People preferred and sought out this kind of HCW when their baby was sick.*" (31 years old caregiver, IDI)

Respondents also emphasized the need to educate the community and raise awareness among neighbors and HCWs who can help mothers identify SAM and encourage fathers to seek early treatment before the illness gets serious. Participants proposed several ideas to strengthen community-based management of children with SAM: expanding the service

delivery facilities to make them easily accessible and making doctors and qualified HCWs more available so that care can be cheaper and more accessible. They also wanted the government to work on reducing poverty so that no babies have not born with malnutrition: "*The government should set policies and implement them to end poverty*." All participants agreed that the SAM service should be free.

**Sub-theme 2:—Perceived causes of SAM and its management.** The perceived causes of a child's SAM were similar among most caregivers and HCWs. These include underlying illnesses, poverty, and local superstitions or taboos. Poor awareness of appropriate feeding, illiteracy, and poor breastfeeding practices were almost unanimously mentioned as major causes of SAM. A few respondents also mentioned workload, leaving home for long periods due to office work, and poor personal hygiene. If the pregnant mother does not get a nutritious diet, then the baby will be malnourished too. If the baby is not getting breastfed properly and the cleanliness of the baby is not maintained properly, then the baby will suffer malnutrition, which may become severe later. The mother needs to get more nutritious foods to keep the baby healthy.

Regarding inappropriate feeding, respondents mentioned that, due to a lack of awareness, some mothers do not breastfeed their babies regularly enough to ensure adequate intake. Instead, they give infant formula or cow's milk when the infant is hungry. As a result, infants would often become ill and hence end up with SAM.

"*Lack of awareness and knowledge on nutrition, especially about child feeding practices, could be the main reasons for malnutrition*". (34 years old health care worker, KII)

The majority of respondents stated that a malnourished mother will give birth to a malnourished child. They identified several causes of maternal malnutrition, including a lack of knowledge about nutritious foods and insufficient care-seeking during pregnancy. If the mother of the baby is malnourished, then the baby can be malnourished too.

A few respondents also stated this problem might be intestinal parasites.

"*To give the child anti helminthiasis to treat intestinal parasites, and give the family food frequently*" (32 years old caregiver, IDIs)

**Sub-theme 3:—The health care context (logistics and trained manpower).** All participants mentioned that logistics are the crucial element for SAM treatment and stated this as one major difficulty.

"*Most drugs and supplies were available in the hospital, but not all of the time, which forced me to buy medicines in a private pharmacy with a costly price* ". (30 years old mother, IDI).

A mother stated:—"*Since there is no television, which aids in health education and prevents boredom because staying too long here could result in depression, and the bed is too small, making it impossible to sleep with our infant, the service delivery needs to be enhanced even more.*". (28 years old mother, IDI)

"*Nowadays the logistics issue is the major problem in most public hospitals for example there is a scarcity of disposable gloves in the hospital, and clients are obliged to purchase them rather than giving with free of charge. Therefore, because of logistics issues, some materials important for SAM management are not fully available in the hospital*". (31 years old health care manager, IDI)

We observed practices and procedures, particularly milk preparation, supplies and equipment for emergency treatment, counseling, guidance, and education, caregiver involvement, the adequacy of rooms and beds, facilities that aid in hygiene and sanitation, the attractiveness of the unit, and whether the unit is separately structured, using the semi-structured observation checklist adopted from the updated national SAM management guideline (2019). The SC unit was separated from other patients' wards within two rooms for treatment of different phases one for phase 1 and the other for phase 2 and about 12 beds were assigned. In the department with O1 MV, there is also an ICU unit for critically ill children that is painted, clean, and odorless and has water and electrical supplies, toilets, showers, and a playroom with toys and carts for the child to use. The SC unit has a separate milk preparation and nursing station, with a physician's duty room near the unit.

The adherence to the STG (Standard treatment guidelines) by care healthcare providers was substandard, especially the feeding protocol, which was not as per the guideline. Therapeutic multi-charts were not used. There are no media resources or job aids available to assist with counseling and health education. The healthcare workers were cooperative. Most emergency drugs were available, but there was a shortage of some equipment's, such as glucose meter strips, a nebulizer, and perfusion, which are critical during management of medical complications. Reading materials and guidelines were available. There are utensils for milk preparation, but no direct water source in this room. The preparation followed the guidelines, but they fed the children four hours per day, rather than two hours per day recommended by the guidelines. In addition, it is difficult for mothers and caregivers to sleep because the beds are too small and uncomfortable, and there is no solution designed to address this issue.

However, in the case of infants, they were not sure that MUAC was applied, so they focused on the visible wasting of arms, legs, and chest. Most of the HCWs also reported that community members, especially parents, were unable to recognize a child or infant with SAM. They said that most cases were identified by HCWs themselves as a secondary finding when caregivers presented a sick child to health facilities.

In most cases, they do not come with SAM as a problem." *They don't usually arrive with SAM as an issue. Only those who have prior experience receiving treatment for this illness can quickly recognize it. Otherwise, in most cases, they come with illness, and then SAM is identified".* (38 years old health care worker, KII)

The SAM diagnosis was not as per the guideline. They used visible, severe wasting as an admission criterion, although the protocol doesn't allow it. The case management and feeding practices of healthcare workers were not based on the recently revised guidelines. Rather than feeding children every two hours during the first day of admission and every three hours during the second day of admission, they gave and prepared milk every four hours. They did not begin the transition and phase 2 with RUTF; instead, they used the F-100 formula, which required them to stay in the hospital for extended periods. They also frequently used MUAC instead of Z-score as a discharge criterion. They did not use the therapeutic multi-chart for close monitoring of most children. The HCW also fails to conduct nutritional counseling but it is mostly focused on advice. Moreover, even though it is expected to have adequately trained HCWs, there is a lack of training.

## Major theme 2:—Facilitators of severe acute malnutrition management

**Sub theme 1:—Able to identify severe acute malnutrition.** Most respondents, including caregivers, were aware of "Amenmin (አመንጓምን)," the local term for "severe acute malnutrition." However, determining whether a child was malnourished or not differed among caregivers and HCWs. Caregivers focused only on physical symptoms and observable signs; HCWs

also mentioned diagnostic tools. Signs and symptoms mentioned by both groups included visible wasting, an inability to feed or breastfeed, and associated illnesses. The baby's appearance was also mentioned; most respondents, particularly caregivers, described a malnourished child as one who is "skinny or bony with a big head, swollen belly, and sunken eyes." A few respondents, particularly the HCWs, mentioned swollen hands and legs or loose skin as additional signs of malnutrition.

> "*The bones of the chest will be visible." His or her eyes will be prominent, the eyes will be pale, there will be less blood in the eye, and he or she will get skinny*". (32 years old health care worker, KII).

> "*When a baby does not receive a healthy diet and, after eating, experiences diarrhea and vomiting, nothing can remain in the stomach for an extended time period, causing the infant to become thinner, more susceptible to colds and fevers, and ultimately developing SAM.* ". (30 years old mother, IDI).

Almost all the HCWs mentioned that they could distinguish different severities of acute malnutrition in children under 5 years old using mid-upper-arm circumference (MUAC) and a z-score. Almost all respondents mentioned the need to visit doctors or other health care providers when the above-mentioned signs are apparent. However, the choice of service provider varied. Most caregivers stated that they had first taken their babies to nearby health care providers and received advice on frequent breastfeeding, the use of diversified additional nutritious foods, such as milk, eggs, a mix of cereals and legumes, potatoes, and oranges, for the child after the age of six months. Some of the caregivers also stated the importance of adequate nutrition for pregnant and breastfeeding mothers.

**Sub theme 2: Service delivery.**   Most respondents and caregivers mentioned that the service delivery was very good. The health care workers were very helpful, caring, and friendly, they involve caregivers involved caregivers in the care and treatment of SAM.

> "*When health care workers told me that my child's problem was SAM, I was perplexed. Because I fed him as much as he needed with our family. Then they respond politely, saying that this is not adequate to feed children with family meals, which is not enough to build their bodies and develop their brains too. After a thorough examination, the health care workers sent me to this hospital with a referral slip. Then, soon after arriving at this Felege Hiwot Comprehensive specialized hospital, healthcare workers examined him well, and lab tests were done. After that, they ordered me to admit him to the ward for nutritional rehabilitation therapy started some injections in his blood vessels, and gave him milk in a red cup, then a blue cup, and now a green cup. Findings were communicated to me. Since the day of admission, my child has gotten better and better. A daily round is conducted; his weight is measured. Now my child is alert, smiley, and even playing, as you saw him at the play area. After 2 months of staying in this hospital, my son is getting better." "Thanks to God and health care workers,*" (34 years old mother, IDI)

Another caregiver mentioned the service provision as "*We are missing nothing from this hospital, and they are all things. In the morning, they give us a lesson on how to wash cups for receiving therapeutic milk, wash ourselves and our child, and make sure there is enough water supply, a toilet, a play area, and even that the walls are painted well and are fine. Linen is provided for changing during the day and night. I lost nothing in the delivery of the service. When I get discharged from this hospital, I will tell and teach others to come here and get treatment for*

*SAM. God bless you. The cleaners, nurses, and doctors are paying for everything for us. We learned a lot for ourselves as well as our children. I did not ask to buy drugs out of the hospital; I got them in the hospital pharmacy. "Even at the beginning of the treatment, my child was refusing to take his therapeutic milk orally, and they inserted a tube for feeding via his nose, and when I got frightened due to the procedure, they smoothly counseled me, i.e., they involved us in the care of the child courageously, and this is what makes me very happy."* (28 years old caregiver, IDI)

The majority of respondents also thought the facility layout was excellent. "*SAM management is so good that it is very comfortable for us because one of them can keep our children and our hygiene, have a toilet, a different room, and there are very pleasant letters and pictures; they are very good; they are very different and better than our home; they gave milk. "They are friendly and familiar. "The respectful workers in the department make me very happy.*" (26 years old caregiver, IDI)

"*I've known this hospital for a few years, and it's improving, not only in terms of infrastructure and green space but also because there are very courageous and compassionate workers, which greatly influenced my decision to continue with SAM treatment for my child." "There is also a playroom with toys and a wall painted with colorful pictures, which makes children happy and friendly.*" (31 years old caregiver, IDI)

"*Healthcare workers are friendly, and they educate, advise, and encourage us daily." God bless them and keep them safe." And "the respectful and courageous approach of doctors and nurses makes me more satisfied and motivated.*" (24 years old caregiver, IDI)

"*The hospital's everyday services are being enhanced. Previously, the hospital lacked an autonomous SAM ward, but now it has one to treat children who are malnourished. Overall, there has been a noticeable improvement in hospital services.*". (40 years old health care manager, KII)

**Sub-theme 3: Participation in community-based health insurance.** Being a member of CBHI increased health-seeking behavior in the majority of respondents. This is supported by the following quotes:

"*I'm a member of CBHI, and it reduces the cost that I could pay and enables me to come here*". (32 years old caregiver, IDI)

" *Being a member of CBHI is crucial, particularly for my child." I wouldn't be able to afford the treatment if I wasn't a member of CBHI.*". (24 years old caregiver, IDI)

## Discussion

This study explores important concepts focusing on SAM management barriers and facilitators among children with SAM. The study identified two major themes and six subthemes.

Signs and symptoms mentioned by caregivers included visible wasting, an inability to feed or breastfeed, and associated illnesses. Most respondents, particularly caregivers, described a malnourished child as one who is "skinny or bony with a big head, swollen belly, and sunken eyes." A few respondents, particularly the HCWs, mentioned swollen hands and legs or loose skin as additional signs of malnutrition. Associated illnesses commonly reported include diarrhea, vomiting, fever, common cold, and cough. Inability to feed or breastfeed, becoming inactive, and crying excessively were additional concerns raised by many caregivers and HCWs. This is similar to a study conducted in Bangladesh and India. This might be due to the study

participants in both studies have relatively similar levels of health literacy and socioeconomic conditions [7,11].

The perception of an "evil eye" was also cited as a reason for the program's low attendance. Mothers showed that there was a lack of knowledge about GMP (Growth Monitoring Program), limited community conversation, and weak counseling about child nutrition as a GMP program explored reasons for low attendance. This finding was not congruent with a study done in Nepal [5]. This may be due to the effectiveness of GMP services was being undermined by underdeveloped health systems and inconsistent implementation. Other reasons mentioned by the mothers were the consideration of the "evil eye" and the method of weighing a child. Further research is needed to explore the implementation of GMP by healthcare workers and to evaluate the extent of the identified reasons for low attendance to the GMP program by the mother [3,23].

Respondents revealed that due to a lack of adequate knowledge of appropriate childhood feeding practices, some mothers do not feed their babies regularly enough to ensure adequate intake. This finding was in line with a study done in Damot Pulassa, Wolaita, South Ethiopia, and Tigray, Ethiopia [24,25]. This may be due to both study areas' participants have relatively similar socio-economic status and literacy levels.

In this study training was mentioned as one key component for SAM management for both health care workers and care givers. This finding was in line with a study done in UK [26]. This may be due to training improves the individual's level of awareness, increases an individual's skill in one or more areas of expertise and increases an individual's motivation to perform their job well.

In this study logistics, staff inadequacy, and other factors were barriers for SAM management. This finding was similar to a study done in South Africa[17] revealed that effective management of pediatrics SAM was affected by a multiplicity of factors that manifest at different levels of the health system, including organizational and system-wide policy and governance challenges; poor strategic staffing mechanisms; supply chain management issues; inadequate skilled medical personnel; poorly managed SAM comorbidities at baseline; misdiagnosis of SAM and related comorbidities at the first point of care; traditional medicine as the first point of care; and prolonged gross negligence of SAM cases by their caregivers. This may be due to the prevalence of undernutrition in children under 5 years of age remains particularly high in Africa and South East Asia.

All participants agreed that logistics are critical for SAM treatment and identified as a major challenge; this finding is consistent with the findings of the Gozamin District [27]. On the contrary, this result differs from the study conducted in Zimbabwe [28]. This difference might be due to the difference in socio-economic status of Zimbabwe and Ethiopia.

The perceived causes of a child's SAM were similar among most caregivers and HCWs. These include underlying illnesses, poverty, and local superstitions or taboos. Poor awareness of appropriate feeding, illiteracy, and poor breastfeeding practices were almost unanimously mentioned as major causes of SAM, which has similar findings in studies conducted in different regions [28–32].

In this study poverty was mentioned as predisposing factor for severe acute malnutrition. This finding was in line with a study done in Uganda [33]. This may be due to relatively similar socio-economic status of the study participants of Uganda and Ethiopia. Participants proposed several ideas to strengthen community-based management of children with SAM. This finding was in line with a study done in Odisha, India [34]. The reason may be due to childhood malnutrition is one of the major public health challenges in India like Ethiopia.

The majority of respondents stated that a malnourished mother will give birth to a malnourished child. They related several reasons for maternal malnutrition, including a lack of

awareness of nutritious food and inadequate care-seeking behavior during pregnancy. If the mother of the baby is malnourished, then the baby can be malnourished too. Almost all the respondents, including caregivers and HCWs, mentioned maternal nutrition and well-being were viewed as essential for the child's well-being [3,29,35–37].

Health care workers in the study setting mentioned that the increased workload, while admissions increased to the malnutrition ward affected the quality of care, staff levels are not always adequate to provide the care they want to provide, which is similar to studies conducted in India and Bangladesh and others [3,8,38–40].

Most caregivers responded that they could not identify/notice SAM until it was severe, this finding was similar to a study done in Bangladesh [11]. This may be due to Bangladesh was among the countries with the highest burden of malnutrition like Ethiopia.

Most respondents identified mothers as having primary caring responsibility for a sick infant. Mothers were said to be best placed to determine the reason why their children were ill and to care for them at home. Fathers and in-laws were also said to be crucial in cases of severe illness when an infant needs to be taken to the hospital. Respondents also emphasized the need to educate the community and raise awareness among neighbors and HCWs who can help mothers identify SAM and encourage fathers to seek early treatment before the illness gets serious [7,23,41–44].

Participants proposed several ideas to strengthen community-based management of children with SAM, expanding service delivery facilities to make them more accessible and increasing the availability of doctors and qualified HCWs to make care more affordable and accessible. They also wanted the government to work on reducing poverty so that no babies are born with malnutrition. The government should set policies and implement them to end poverty. All participants agreed that SAM services, like maternity services, should be free of charge. Most respondents and caretakers mentioned that the service delivery was very good. The health team is very helpful, caring, and friendly; they involve caregivers in the care and treatment of SAM. The majority of respondents also thought the facility layout was excellent. This may be due to the flagship agendas of CRC and CASH by the Ministry of Health in Ethiopia, which contrasts a study finding in the Loka Abaya district that limited community conversation and weak counseling about child nutrition as a GMP program were explored as reasons for low attendance [45].

## Strength and limitation

The strength of the study was the sampling method and design which is a criterion-based, heterogeneous purposive sampling technique and using descriptive phenomenological study design and data triangulation both observation, in-depth and key informant interview. In addition, the study used different healthcare workers and healthcare managers to reveal several views. Moreover, it was the first study in the country and the study area on the issue.

Findings may be subject to social desirability bias; however, they offer an understanding of the complexity and dynamics of different barriers and facilitators that contributed to and interacted with the management of SAM by the mothers, HCWs, and HCMs. Since the study was phenomenological study design researcher-induced bias can affect the outcome of the study but we try to reduce it. Moreover, the study used purposive sampling, which might not represent the whole population or lack generalizability.

## Implication of the study

First, we note that both facilitators and barriers are revealed as significant influences. Various stakeholders may advocate for action on particular barriers or facilitators. Provide support to a

sustainable intervention and foundation to implement social and behavior change. Under-standing and exploring the barriers, and facilitators on the management of SAM among mothers/caregivers, healthcare providers, and healthcare managers provide an additional body of knowledge, to see the extent of the problem and to develop intervention tools and strategies to improve SAM treatment.

## Conclusion

Effective management of SAM was affected by a multiplicity of factors. The results reaffirm how socioeconomic and sociocultural conditions, perceived causes of SAM and its management and the health care context were the major barriers, while identification of SAM, service provision, and being a member of CBHI were the major facilitators for SAM management. Hence, multiplicity of factors should be considered to successfully implement the program. Therefore, special attention shall be given to SAM management.

## Supporting information

**S1 File. Interview questions for the study.**
(PDF)

## Acknowledgments

We would like to express our heartfelt gratitude to Bahir Dar University College of Medicine and Health Sciences, Bahir Dar City Administration, Felege Hiwot Comprehensive Specialized Hospital, especially the pediatric ward, and the participants of the study who gave important information for this research.

## Author Contributions

**Conceptualization:** Daniel Alelign.

**Formal analysis:** Zeamanuel Anteneh Yigzaw.

**Methodology:** Netsanet Fentahun, Zeamanuel Anteneh Yigzaw.

**Software:** Netsanet Fentahun.

**Writing – original draft:** Daniel Alelign.

**Writing – review & editing:** Zeamanuel Anteneh Yigzaw.

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
