## [Decision Letter · Decision Letter 0]

19 Sep 2023

PONE-D-23-21702Barriers and facilitators of severe acute malnutrition management at Felege Hiwot comprehensive specialized hospital, a descriptive phenomenological study, North West Ethiopia, Bahir Dar, 2022.PLOS ONE

Dear Dr. yigzaw,

Thank you for submitting your manuscript to PLOS ONE. After careful consideration, we feel that it has merit but does not fully meet PLOS ONE’s publication criteria as it currently stands. Therefore, we invite you to submit a revised version of the manuscript that addresses the points raised during the review process.

Please address the comments forwarded from both reviewersSorry for the time we spent to provide feedback, it is because of the challenges to find reviewrs.==============================

We look forward to receiving your revised manuscript.

Kind regards,

Kahsu Gebrekirstos Gebrekidan

Academic Editor

PLOS ONE

Journal Requirements:

"No funding at all"

"No authors have competing interests"

Reviewers' comments:

Reviewer's Responses to Questions

**Comments to the Author**

1. Is the manuscript technically sound, and do the data support the conclusions?

Reviewer #1: Partly

Reviewer #2: Yes

2. Has the statistical analysis been performed appropriately and rigorously? 

Reviewer #1: No

Reviewer #2: Yes

3. Have the authors made all data underlying the findings in their manuscript fully available?

Reviewer #1: Yes

Reviewer #2: Yes

4. Is the manuscript presented in an intelligible fashion and written in standard English?

Reviewer #1: No

Reviewer #2: Yes

5. Review Comments to the Author

Reviewer #1: The research area is very alarming. However, choosing the study area is not in line with current problem of the country and the region. I recommend including rural areas as well as urban areas for better understanding of barriers and facilitators of severe acute malnutrition management.

Why chose exploratory Phenomenology qualitive research?

How do you came up with samples (9,3,2)?

Why the PI only did the thematic analysis? How you did you control bias?

Reviewer #2: 1.Rigor of the study is indicated Through triangulations i.e. Data source triangulation; Method triangulation(observation and qualitative).In order to make the work free of bias, the author did not seem triangulate via investigators triangulation as well.

2.The full name of the Atlas-Ti software , year and the exact version number should me mentioned

3.How did service providers recognize the 2 types of SAM, while this could be facilitator/barrier for the treatment.

4.References are not checked and corrected such as the Author surname, Article title, Journal Name, Volume(Issue), pp. page range. It seems the author relied on citation software with out cross edition. For e.g:2,5,7,11,17,23,34 and more

6. PLOS authors have the option to publish the peer review history of their article (what does this mean?). If published, this will include your full peer review and any attached files.

Reviewer #1: No

Reviewer #2: **Yes: **beyene Wondafrash

---

## [Author Response · Author response to Decision Letter 0]

28 Oct 2023

Dear Reviewers

Thank you very much for your valuable comments.

As per your comments. We try to amend it or do major revisions to the points you raised and other issues to be amended. we addresses all the comments raised.

Thank you very much for your support

---

## [Decision Letter · Decision Letter 1]

26 Dec 2023

PONE-D-23-21702R1Barriers and facilitators of severe acute malnutrition management at Felege Hiwot Comprehensive Specialized Hospital, North West Ethiopia, Bahir Dar, a descriptive phenomenological studyPLOS ONE

Dear Dr. yigzaw,

Thank you for submitting your manuscript to PLOS ONE. After careful consideration, we feel that it has merit but does not fully meet PLOS ONE’s publication criteria as it currently stands. Therefore, we invite you to submit a revised version of the manuscript that addresses the points raised during the review process.

We look forward to receiving your revised manuscript.

Kind regards,

Kahsu Gebrekidan

Academic Editor

PLOS ONE

Additional Editor Comments :

The decision is major revision again because the second reviewer did not respond so that we invited new reviewer. As a result, it took us a lot of time.

Reviewers' comments:

Reviewer's Responses to Questions

**Comments to the Author**

1. If the authors have adequately addressed your comments raised in a previous round of review and you feel that this manuscript is now acceptable for publication, you may indicate that here to bypass the “Comments to the Author” section, enter your conflict of interest statement in the “Confidential to Editor” section, and submit your "Accept" recommendation.

Reviewer #2: All comments have been addressed

Reviewer #3: (No Response)

2. Is the manuscript technically sound, and do the data support the conclusions?

Reviewer #2: Yes

Reviewer #3: No

3. Has the statistical analysis been performed appropriately and rigorously? 

Reviewer #2: Yes

Reviewer #3: No

4. Have the authors made all data underlying the findings in their manuscript fully available?

Reviewer #2: Yes

Reviewer #3: No

5. Is the manuscript presented in an intelligible fashion and written in standard English?

Reviewer #2: Yes

Reviewer #3: No

6. Review Comments to the Author

Reviewer #2: This research work will help researchers to understand pure qualitative approach. It will be good if such a research could include different centers (multicenter or Rural or urban) with adequate sample.

Reviewer #3: Dear authors

I would like to express my sincere appreciation for your dedicated efforts in conducting the research titled "Barriers and facilitators of severe acute malnutrition management at Felege Hiwot Comprehensive Specialized Hospital, North West Ethiopia, Bahir Dar, a descriptive phenomenological study" submitted to PLOS ONE under the identifier PONE-D-23-21702R1

In general, the study is required however,

- Rigor of the study is very weak and questionable as researchers based their results on mere self-reported data from 15 participants. This is the big issue in this study since validity and credibility of results are questionable and not convincing.

- Using the phenomenological design is unclear and require further justifications

- Sampling is not convincing and need more justifications

- Themes are vague and cannot reflect the results

- Conclusions cannot drawn from the study

- Background is very weak and need to be strengthened in the light of international studies

- You should generate the research gap for this study

- You should follow guidelines of scientific writing

- Discussion does not follow guidelines of effective debate. It require in-depth analysis and more criticism

- What about the implications of this study. How this study could add for clinicians and heath care management. How this study could shape heath policy in Ethiopia?. You should add a separate section for implications

- The majority of references cited are outdated ( more than five years )

-

Thanks

7. PLOS authors have the option to publish the peer review history of their article (what does this mean?). If published, this will include your full peer review and any attached files.

Reviewer #2: **Yes: **Dr beyene Wondafrash Ademe

Reviewer #3: No

---

## [Author Response · Author response to Decision Letter 1]

9 Jan 2024

Dear Reviewers

Thank you very much for your support.

We try to do a good paper with high public health importance. This study was not done as such in higher numbers over all the world. It was the first study especially in Ethiopia with a descriptive phenomenological study design. 

We address all the points commented on line by line. Moreover, we add some points for the better of the manuscript.

Now it is a good paper. Even we are eager to amend it if you have any amendments required or any comments there. But we see it Through word by word and line by line, now it is very SMART paper.

Plos One is very ideal journal for this qualitative research, we are eager to publish other incoming papers also in this prestigious journal.

We think that Now It is a very excellent paper and of great public health importance. 

Thank You very much for your consideration for publication.

Thank You

---

## [Decision Letter · Decision Letter 2]

22 Jan 2024

PONE-D-23-21702R2Barriers and facilitators of severe acute malnutrition management at Felege Hiwot Comprehensive Specialized Hospital, North West Ethiopia, Bahir Dar, a descriptive phenomenological studyPLOS ONE

Dear Dr Zeamanuel,

Thank you for submitting your manuscript to PLOS ONE. After careful consideration, we feel that it has merit but does not fully meet PLOS ONE’s publication criteria as it currently stands. Therefore, we invite you to submit a revised version of the manuscript that addresses the points raised during the review process. Please submit your revised manuscript by Mar 07 2024 11:59PM. If you will need more time than this to complete your revisions, please reply to this message or contact the journal office at plosone@plos.org. Please include the following items when submitting your revised manuscript:A rebuttal letter that responds to each point raised by the academic editor and reviewer(s). You should upload this letter as a separate file labeled 'Response to Reviewers'.A marked-up copy of your manuscript that highlights changes made to the original version. You should upload this as a separate file labeled 'Revised Manuscript with Track Changes'.An unmarked version of your revised paper without tracked changes. You should upload this as a separate file labeled 'Manuscript'.

We look forward to receiving your revised manuscript.

Kind regards,

Kahsu Gebrekidan

Academic Editor

PLOS ONE

Reviewers' comments:

Reviewer's Responses to Questions

**Comments to the Author**

1. If the authors have adequately addressed your comments raised in a previous round of review and you feel that this manuscript is now acceptable for publication, you may indicate that here to bypass the “Comments to the Author” section, enter your conflict of interest statement in the “Confidential to Editor” section, and submit your "Accept" recommendation.

Reviewer #3: (No Response)

2. Is the manuscript technically sound, and do the data support the conclusions?

Reviewer #3: Partly

3. Has the statistical analysis been performed appropriately and rigorously? 

Reviewer #3: No

4. Have the authors made all data underlying the findings in their manuscript fully available?

Reviewer #3: No

5. Is the manuscript presented in an intelligible fashion and written in standard English?

Reviewer #3: No

6. Review Comments to the Author

Reviewer #3: Dear esteemed authors

I am pleased to acknowledge the improvement made to the manuscript. However, comments on original paper need to be addressed.

- Conclusion cannot be drawn from this study.

- Data regarding rigor, sampling and design are still not convincing at this point. The issue of reaching to data saturation from self-reporting of 15 subjects is very questionable

- Theoretical underpinnings of the study is still weak

- Results are vague and need to be re written in clear and informative manner

- Discussion of findings is not adequately explained

Best regards

7. PLOS authors have the option to publish the peer review history of their article (what does this mean?). If published, this will include your full peer review and any attached files.

Reviewer #3: No

---

## [Author Response · Author response to Decision Letter 2]

31 Jan 2024

Dear Reviewers

Greetings.

Thank you very much for your support

We do very SMART qualitative research with great public health importance. The research is the first study in the study area and Ethiopia. We address all the points raised by the reviewer; some sentences may be similar to the previous one the reason was we believe that we do best on that as it is. We cite more than 45 references. We do the research with a legal research ethical letter (attached). We try to rewrite the result and the conclusion part as per your comment. If you have other review comments, we are ready to amend them again.

Plos one was ideal for this qualitative research.

We hope we are waiting for this publication in this journal.

Thank you very much for your support and understanding.

---

## [Decision Letter · Decision Letter 3]

13 Feb 2024

Barriers and facilitators of severe acute malnutrition management at Felege Hiwot Comprehensive Specialized Hospital, North West Ethiopia, Bahir Dar, a descriptive phenomenological study

PONE-D-23-21702R3

Dear Mr Zeamanuel,

We’re pleased to inform you that your manuscript has been judged scientifically suitable for publication and will be formally accepted for publication once it meets all outstanding technical requirements.

Kind regards,

Kahsu Gebrekidan

Academic Editor

PLOS ONE

Additional Editor Comments (optional):

Reviewers' comments:

Reviewer's Responses to Questions

**Comments to the Author**

1. If the authors have adequately addressed your comments raised in a previous round of review and you feel that this manuscript is now acceptable for publication, you may indicate that here to bypass the “Comments to the Author” section, enter your conflict of interest statement in the “Confidential to Editor” section, and submit your "Accept" recommendation.

Reviewer #3: (No Response)

2. Is the manuscript technically sound, and do the data support the conclusions?

Reviewer #3: Partly

3. Has the statistical analysis been performed appropriately and rigorously? 

Reviewer #3: Yes

4. Have the authors made all data underlying the findings in their manuscript fully available?

Reviewer #3: Yes

5. Is the manuscript presented in an intelligible fashion and written in standard English?

Reviewer #3: Yes

6. Review Comments to the Author

Reviewer #3: Dear respected professor editor

Dear respected authors

Thank you for this opportunity

The manuscript demonstrate much improvement at this point.

It could be published after making a through editing for English use and grammar. Also, implications need to be strengthened to give clear insight about how your study add to literature and practice .

Best wishes

7. PLOS authors have the option to publish the peer review history of their article (what does this mean?). If published, this will include your full peer review and any attached files.

Reviewer #3: **Yes: **Dr.Ahmed Abdelwahab

---

## [Editor Report · Acceptance letter]

26 Feb 2024

PONE-D-23-21702R3 

PLOS ONE

Dear Dr. Yigzaw, 

I'm pleased to inform you that your manuscript has been deemed suitable for publication in PLOS ONE. Congratulations! Your manuscript is now being handed over to our production team.

Kind regards, 

on behalf of

Dr. Kahsu Gebrekidan 

Academic Editor

PLOS ONE